**Data Availability Statement:** RNA sequence files have been deposited in the NCBI Sequence Read

# Pathways of calcium regulation, electron transport, and mitochondrial protein translation are molecular signatures of susceptibility to recurrent exertional rhabdomyolysis in Thoroughbred racehorses

Kennedy Aldrich[1], Deborah Velez-Irizarry[1], Clara Fenger[2], Melissa Schott[1], Stephanie J. Valberg[1]*

**1** Mary Anne McPhail Equine Performance Center, Large Animal Clinical Sciences, College of Veterinary Medicine, Michigan State University, East Lansing, MI, United States of America, **2** Equine Integrated Medicine, PLC, Lexington, KY, United States of America

☯ These authors contributed equally to this work.

* valbergs@cvm.msu.edu

## Abstract

Recurrent exertional rhabdomyolysis (RER) is a chronic muscle disorder of unknown etiology in racehorses. A potential role of intramuscular calcium ($Ca^{2+}$) dysregulation in RER has led to the use of dantrolene to prevent episodes of rhabdomyolysis. We examined differentially expressed proteins (DEP) and gene transcripts (DEG) in gluteal muscle of Thoroughbred race-trained mares after exercise among three groups of 5 horses each; 1) horses susceptible to, but not currently experiencing rhabdomyolysis, 2) healthy horses with no history of RER (control), 3) RER-susceptible horses treated with dantrolene pre-exercise (RER-D). Tandem mass tag LC/MS/MS quantitative proteomics and RNA-seq analysis (FDR <0.05) was followed by gene ontology (GO) and semantic similarity of enrichment terms. Of the 375 proteins expressed, 125 were DEP in RER-susceptible versus control, with 52 ↑DEP mainly involving $Ca^{2+}$ regulation (N = 11) (e.g. RYR1, calmodulin, calsequestrin, calpain), protein degradation (N = 6), antioxidants (N = 4), plasma membranes (N = 3), glyco(geno)lysis (N = 3) and 21 DEP being blood-borne. ↓DEP (N = 73) were largely mitochondrial (N = 45) impacting the electron transport system (28), enzymes (6), heat shock proteins (4), and contractile proteins (12) including $Ca^{2+}$ binding proteins. There were 812 DEG in RER-susceptible versus control involving the electron transfer system, the mitochondrial transcription/translational response and notably the pro-apoptotic $Ca^{2+}$-activated mitochondrial membrane transition pore (*SLC25A27*, *BAX*, *ATP5* subunits). Upregulated mitochondrial DEG frequently had downregulation of their encoded DEP with semantic similarities highlighting signaling mechanisms regulating mitochondrial protein translation. RER-susceptible horses treated with dantrolene, which slows sarcoplasmic reticulum $Ca^{2+}$ release, showed no DEG compared to control horses. We conclude that RER-susceptibility is associated with alterations in proteins, genes and pathways impacting myoplasmic $Ca^{2+}$ regulation, the mitochondrion and protein degradation with opposing effects on

Archive (BioProject PRJNA641844, accession numbers SRR10997329 – SRR10997344). Mass spectrometry proteomic data are available via the ProteomeXchange Consortium PRIDE repository (http://www.ebi.ac.uk/pride/archive/projects/PXD020100).

**Funding:** Research was supported by the Grayson Jockey Club Research Foundation (SJV), the Mary Anne McPhail Endowment at Michigan State University (SJV) and in part by NIH Grant T32 OD011167 to Michigan State University (KJA). The Grayson Jockey Club Research Foundation provided funds to perform the research study and partial salary for one author (DVI). CF is the owner of Equine Integrated Medicine PLC. and identified horses in her practice that were healthy controls or susceptible to RER and facilitated biopsy samples that were obtained by another author (SJV). CF did not receive any financial remuneration for participating in the study. The specific role of these authors is articulated in the 'author contributions' section. The funders had no role in study design, data collection and analysis, decision to publish, or preparation of the manuscript.

**Competing interests:** One of the authors (CF) is a practicing veterinarian and owner of the business Equine Integrated Medicine PLC. There are no other relevant declarations relating to employment, consultancy, patents, products in development, or marketed products. This does not alter our adherence to PLOS ONE policies on sharing data and materials.

mitochondrial transcriptional/translational responses and mitochondrial protein content. RER could potentially arise from excessive sarcoplasmic reticulum $Ca^{2+}$ release and subsequent mitochondrial buffering of excessive myoplasmic $Ca^{2+}$.

## Introduction

Recurrent exertional rhabdomyolysis (RER) develops in 5–10% of Thoroughbred racehorses, with intermittent episodes frequent enough in one third of RER susceptible horses to disrupt a regular training and racing schedule [1,2]. Both environmental factors and a genetic predisposition impact the expression of RER with young, nervous, female horses in race training on high starch diets affected at highest frequencies [1,3,4]. Although heritability of RER is estimated at 0.40, linkage analyses and genome-wide association studies so far have been ineffective at identifying a consensus chromosomal locus or gene associated with RER in Thoroughbred horses [5–7]. This could be due to a polygenic mode of inheritance or post-translational modifications that arise in a stressful training environment.

Several studies have implicated an abnormality in intramuscular $Ca^{2+}$ regulation as the underlying basis for RER [8–12]. Methodologies used to study RER have included caffeine-induced contractures in muscle bundles and muscle cell cultures, $Ca^{2+}$ release in isolated sarcoplasmic reticulum membranes (SR) and microarray studies of gene expression [8–12]. Based on a potential dysregulation of $Ca^{2+}$ release with RER, dantrolene, an inhibitor of the sarcoplasmic reticulum $Ca^{2+}$ release channel (RYR1), has been used prophylactically prior to exercise to decrease muscle degeneration in RER-susceptible Thoroughbreds at doses between approximately 0.8–4 mg/kg [13–16]. Side effects of dantrolene, including decreased cardiac output and hyperkalemia, are reported in horses when higher doses (6 mg/kg) are given prior to anesthesia [16]. In a study of two Standardbred trotters that subsequently developed RER, altered flux through the mitochondrial electron transport system was identified and this was proposed to have a role in the etiology of RER [17]. Thus, both abnormalities in intramuscular $Ca^{2+}$ regulation and mitochondrial electron transport have been suggested to cause RER, however, the specific basis for RER is not well understood.

The recent application of RNA-seq transcriptomic analysis combined with tandem mass tag liquid chromatography mass spectrometry (TMT LC/MS/MS) quantitative proteomic analyses has provided a global means to study equine myopathies and determine underlying molecular signatures for muscle diseases [18]. With these techniques, genes and proteins can be identified that are expressed in diseased horses at greater or lesser levels than control horses. Gene Ontology (GO) enrichment can then be applied to interpret the genes or proteins that are differentially expressed by assigning them to a set of predefined pathways depending on their functional characteristics. Key molecular signatures of disease can thereafter be identified by evaluating the semantic similarities of these pathways and their relationship to muscle physiology and pathophysiology.

We hypothesize that a combined proteomic and transcriptomic approach utilizing skeletal muscle samples obtained in the same high stress training environment from control and RER mares would identify pathways regulating $Ca^{2+}$ flux as key molecular signatures of RER susceptibility. Muscle samples were taken between episodes of rhabdomyolysis to study molecular signatures of underlying susceptibility rather than signatures of acute rhabdomyolysis. Our first objective was to identify differentially expressed proteins (DEP) and their associated biological pathways in the muscle of RER-susceptible versus control horses. The second objective

was to identify differentially expressed gene transcripts (DEG) and their associated biological pathways in the muscle of RER-susceptible versus control horses as well as DEG in RER-susceptible horses treated with dantrolene prior to exercise (RER-D) versus control horses. The final objective was to integrate the DEP and DEG within significant biological pathways to synthesize a coherent pathophysiologic basis for RER.

## Materials and methods

### Horses

**RER-susceptible horses.** RER-susceptible horses were all Thoroughbred mares housed at one race training center in Lexington KY that were fed 4–5 kg/day of the same concentrate. The concentrate, Race-13, was made by Hallway Feeds (Lexington KY,) and was composed of 8% water soluble sugar, 28% starch and 6% fat. Both the RER-susceptible and control horses were in full race training that typically consisted of the following regime: Monday- trot for 1.6 km, Tuesday through Thursday trot for 1.6 km and slow galloping for 1.6 km, Friday- fast gallop at 80–100% race effort and Sunday- hand walking.

RER horses had a chronic history of exertional rhabdomyolysis with a veterinarian confirming at least two episodes through analysis of serum creatine kinase (CK) and aspartate aminotransferase activities (AST) taken when the horse had clinical signs of muscle pain, stiffness and reluctance to move. RER-susceptible horses were divided into two groups; 1) Untreated RER-susceptible horses which had not received dantrolene in the weeks prior to the study (RER-susceptible, N = 5) and (2) RER-susceptible horses that had a minimum of a one-week history of receiving 0.5 to 1 mg/kg dantrolene administered orally by syringe approximately 2 h before exercise (RER-D, N = 5). The horses' medications were under the control of the individual veterinarians, however, in general, the RER horses that received dantrolene were those that trainers felt were so recurrent that the disorder could not be controlled through less expensive management changes.

**Control horses.** Control horses (N = 5) were Thoroughbred mares in full race training at the same training facility on the same diet that had no history of RER. Histories, plasma CK and AST activities as well as muscle histopathology was assessed to ensure that there was no evidence of current or previous rhabdomyolysis in this group.

### Plasma samples and muscle biopsies

Samples were obtained in the morning between 1 and 5.5 h after exercise with no difference in the time between exercise and biopsy between control and RER susceptible horses (Table 1). In each of the control and RER susceptible groups samples were obtained after; 1 horse trotted

**Table 1. The number of controls, RER-susceptible (no dantrolene) and RER mares treated with dantrolene (RER-D).**

|  | N | Age | Time since last work | Post exercise AST (range) | Post exercise CK (range) | Type 1 | Type 2A | Type 2X |
|---|---|---|---|---|---|---|---|---|
|  |  | yrs | Hours | U/L | U/L | % | % | % |
| **Control** | 5 | 3.0 ± 0.9 | 3.7 ± 1.3 | 359 ± 108 (263–537) | 240 ± 57 (172–342) | 12.6 ± 3.9 | 29.5 ± 14.4 | 58.0 ± 15.2 |
| **RER-susceptible** | 5 | 4.0 ± 1.5 | 2.2 ± 1.2 | 557 ± 305 (219–921) | 349 ± 207 (131–663) | 9.0 ± 2.2 | 36.6 ± 4.9 | 54.4 ± 5.5 |
| **RER-D** | 5 | 3.6 ± 0.8 | 3.1 ± 1.6 | 568 ± 191 (254–840) | 941 ± 828 (200–2445) | 10.1 ± 2.2 | 34.7 ± 10.6 | 55.2 ± 11.7 |
| P value |  | 0.48 | 0.33 | 0.33 | 0.14 | 0.18 | 0.61 | 0.89 |

The mean (SD) age, sex, time lapse between exercise and muscle biopsy, plasma aspartate transaminase (AST) and creatine kinase (CK) activities at time of biopsy, and percentage of gluteal muscle type 1, 2A and 2X fibers are shown. There were no significant differences in any parameters between control, RER-susceptible and RER-D horses.

1.6 km, 3 horses galloped approximately 1.6 km after trotting 1.6 km and one horse was hand walked. For RER-D, samples were obtained after; 3 horses trotted 1.6 km and then galloped approximately 1.6 km and two horses trotted 1.6 km. Participation was at the discretion of the trainers and exercise was determined by the trainers' routine which meant that sampling could not be completely standardized in terms of timing after exercise. Horses had no clinical evidence of muscle pain or stiffness following exercise at the time of sampling. Samples were obtained through a signed consent form and Research was approved by the Institutional Animal Care and Use Committee at Michigan State University number PROTO201900038.

Seven ml blood samples from the jugular vein were collected into heparinized vacutainer tubes, kept on ice packs for up to 6 h, plasma removed after centrifugation and frozen. Plasma CK and AST activities were analyzed within 72 h at the Michigan State University Diagnostic Laboratory. For muscle sampling, horses were sedated with 200 mg xylazine (AnaSed, Santa Cruz Animal Health, Santa Cruz CA) and percutaneous gluteus medius muscle biopsies obtained at a standardized site as previously described [19]. Approximately 200 mg of skeletal muscle was divided into two aliquots, one was immediately frozen in liquid nitrogen and the other was placed in saline moistened gauze and kept on icepacks until frozen in methylbutane suspended in liquid nitrogen within 6 h of sampling. Muscle was stored at -80˚C until analysis.

**Muscle histochemistry.** Eight μm thick sections of muscle were stained with hematoxylin and eosin (HE), modified Gomori Trichrome (GT), nicotinamide adenine dinucleotide tetrazolium reductase (NADH-TR) and adenosine triphosphatase (ATPase, pH 4.4) [20]. Sections were evaluated for the presence of internalized myonuclei, acute necrosis, macrophages, basophilic regenerative fibers and mitochondrial staining pattern. Muscle fiber types were determined using ATPase stains pre-incubated at pH 4.4. Muscle fiber type composition for type 1, 2A, and 2X fibers were determined from approximately 250 muscle fibers per horse.

## Analyses of signalment, plasma CK, AST and fiber composition

The age, timing of biopsy after exercise, plasma CK and AST activities were tested for normality using Kolmogorov-Smirnov test, data that were not normally distributed (age) were log transformed prior to performing a One-way analysis of variance (ANOVA) to compare control, RER-susceptible and RER-D groups. The percentage of type 1, 2A and 2X fibers in a biopsy were compared among the same groups using ANOVA. Significance was set at P <0.05. Data are expressed as mean and standard deviation (SD).

## Proteomics

**Sample preparation and LC/MS/MS.** Proteins were extracted from snap frozen gluteus medius muscle samples (5 RER-susceptible and 5 control) in fresh RIPA lysis buffer (Thermo Scientific, Waltham, MA) with protease inhibitor cocktail (cOmplete, Mini, EDTA-free, Roche). Protein concentration was measured by standard BCA assay (Pierce[TM] Biotechnology, Rockford, IL) and Coomassie-stained SDS gel. LC/MS/MS was performed at the Michigan State University Research and Technology Support Facility (MSU-RTFS) Proteomics Core. In brief, protein samples were subjected to proteolytic digestion with trypsin using the Filter-Aided Sample Preparation protocol (MW cutoff of 30, 000Da). The resulting peptides were labeled with TMT11-131C (Thermo Scientific, Waltham, MA), one sample was run in duplicate as an internal control. Tagged peptides were separated and eluted with the Thermo Acclaim PepMap RSLC column over 125 min at a constant flow rate. Eluted peptides were sprayed into a ThermoScientific Q-Exactive HF-X mass spectrometer (Thermo Scientific, Waltham, MA) using a FlexSpray spray ion source. The top 20 ions in each survey scans (Orbi trap 120,000 resolution at m/z 200) were subjected to higher energy collision induced

dissociation with fragment spectra acquired at 45,000 resolution. The resulting MS/MS spectra were processed with Proteome Discoverer v2.2 (Thermo Scientific, Waltham, MA) and searched against the EquCab3.0 UniProt:UP000002281 protein database using three search engines: Sequest HT, Mascot v2.6 and X! Tandem, to annotate the peptide spectra. Mass spectrometry proteomic data are available via the ProteoeXchange Consortium PRIDE repository with identifier PXD020100 (DIO:10.6019/PXD020100).

**Quantitative data analysis.**  Scaffold Q+ (version Scaffold_4.8.7, Proteome Software Inc., Portland, OR) was used to quantitate TMT-11 plex labeled peptide and protein identifications. Peptide identifications were accepted if they could be established at greater than 95.0% probability. Protein identifications were accepted if they could be established at greater than 99.0% probability and contained at least two identified peptides. A second filter was applied to remove proteins with missing values. Proteins that contained similar peptides and could not be differentiated based on MS/MS analysis alone were grouped to satisfy the principles of parsimony. Proteins sharing significant peptide evidence were grouped into clusters. Spectra data were log-transformed, pruned of those matched to multiple proteins, and weighted by an adaptive intensity weighting algorithm. Of 27,997 spectra in the experiment at the given thresholds, 19,242 (69%) were included in quantitation. Differentially expressed proteins (DEP) between control and RER-susceptible horses were determined by applying Permutation Tests within Scaffold Q+ and a significance threshold of Benjamini-Hochberg FDR≤0.05.

## Transcriptomics

**RNA extraction and sequencing.**  Total muscle RNA was isolated from snap frozen gluteus medius muscle samples as previously described [21]. Sequencing was performed at the MSU-RTSF Genomics Core. Briefly, libraries were constructed per horse with a polyA+ capture protocol using the Illumina TruSeq Stranded mRNA Library Preparation Kit and sequenced on the Illumina HiSeq 4000 platform (2 x 150bp, paired-end reads) [21]. Base calling was done by Illumina Real Time Analysis (RTA) v2.7.7 and the output of RTA was demultiplexed and converted to FastQ format with Illumina Bcl2fastq v2.19.1.

**Mapping and assembling.**  Mapping and assembly were performed as previously described using EquCab 3.0. Alignment statistics and base coverage were obtained with Tophat2 and SAMtools [22]. Total gene transcript expression was quantified for unique sequence reads using HTSeq [23]. Gene transcripts with less than two sequence read counts in at least one horse were removed from further analysis to reduce the number of genes with low expression, leaving 14,155 gene transcripts for differential expression (DE) analyses and gene set enrichment. RNA sequence files have been deposited in the NCBI Sequence Read Archive BioProject PRJNA641844, accession numbers SRR10997329 –SRR10997344.

**RNA-Seq count normalization and transformation.**  The trimmed mean of M-values (TMM) scaling factors [24] were calculated from the weighted mean of $\log_2$ expression ratios using the function calcNormFactors from the Bioconductor R package edgeR. The TMM normalization factors were used to normalize the library sizes of each sample; this step helps removes any systemic technical effects biasing transcript expression between samples. Raw gene transcript abundances were transformed to approximate a normal distribution by calculating the $\log_2$ counts per million (CPM), which is the $\log_2$ of the raw counts and scale-normalized library size ratio. The mean-variance trend of gene transcripts was estimated and incorporated in the variance modeling of the DE analysis as precision weights to accounting for observational level and sample-specific parameters shared across genes [25,26]. The CPM and precision weights were computed using the function voomWithQualityWeights from the Bioconductor R package limma. Principal component (PC) analysis plots were generated for

the CPM gene transcript matrix in R version 3.6.2 using the functions prcomp and ggbiplot [27].

**Differential gene expression analysis.** Differential expression of gene transcripts (DEG) was identified using limma linear models by weighted least squares [28,29]. Linear combinations of model parameters were used to evaluate the DE between control and RER horses. The fixed effects model with $n$ horses and $m$ genes can be represented as:

$$y_{ij} = \mu_j + \beta_j + e_{ij}; i = 1, \ldots n; j = 1, \ldots m$$

Where $y$ is a matrix of CPM with dimension $m \cdot n$, $\mu$ is the overall mean, $\beta$ represents effects due to disease state, and $e$ represents the error term, $e \sim N(0, \sigma_e^2 \, diag(\sqrt{\hat{w}}))$. The heteroscedastic variance was modeled with the estimated precision weights, $\hat{w}_{ij}$. DE was determined based on $t$-tests relative to a threshold (TREAT) where the threshold used in this analysis was a log [30]. Multiple test correction was performed with false discovery rate less than 0.05. q-values, an adjusted or corrected p-value based on estimations of the proportion of false positives, were used in tables to show significance.

**Enrichment and pathway analysis.** Proteins and genes exhibiting significant differential expression between control and RER horses ($FDR \leq 0.05$) were analyzed using the R package clusterProfiler for Gene Ontology (GO) and KEGG pathway enrichment analysis based on the hypergeometric distribution [31,32]. Gene symbols were converted to ENTREZ gene IDs using the human annotation (org.Hs.eg.db: Genome wide annotation for Human. *Bioconductor* R package version 3.7.0) and GO for biological processes, cellular function, and molecular function determined. Pathway enrichment was evaluated separately for the DEP and DEG with negative $\log_2$ FCs (down-regulated) and positive $\log_2$ FCs (up-regulated). The pathways for DEP and DEG that were either significantly upregulated or downregulated relative to background protein/gene expression were determined after multiple test correction with an $FDR \leq 0.05$. The background list used in the pathway analysis consisted of all proteins/genes expressed in the gluteal muscle biopsies and with an equivalent human annotation (11,095 gene profiles, 344 protein profiles). Enriched GO pathways associated with biological processes (BP) were further evaluated for functional similarities between up-regulated and down-regulated DEP and DEG. We computed the semantic similarities between enriched BP identified for DEP and DEG using Jiang and Conrath's Information Content [33] and the R package GOSemSim [34]. This IC-based method estimates the similarity of two GO terms given the annotation statistics of their most informative common ancestor. We used a similarity Conrath's Information Content cutoff of 0.65 to identify dysregulated pathways shared across the transcriptome and proteome of RER horses.

# Results

## Horses and muscle histochemistry

There was no significant difference in age, time between exercise and sampling, plasma CK or AST activities among the control, RER-susceptible, and RER-D horses (Table 1). The range in CK activities was; control 172–342 U/L, RER-susceptible 219–921 U/L and RER-D 185–2445 U/L. There were no histopathologic abnormalities in control horses as part of their inclusion criteria. Two RER-susceptible and three RER-D horses possessed centrally located nuclei in mature myofibers. A few macrophages were present in one RER-D horse. There were no significant differences between RER-susceptible, RER-D and control horses in the percent type 1, type 2A or type 2X fibers (Table 1). ATPase staining could not be obtained in one horse (RER-susceptible) because sections would not remain affixed to the slides during staining.

## Proteomics

**Confident protein classification and differential expression.** Of the 727 proteins quantified by mass spectrometry, 401 had < 0.1% probability of incorrect protein identification and greater than two identified proteins. Filtering for proteins with missing values removed 6.5%, leaving 375 proteins expressed across all horses. The permutation based DE analysis identified 125 DEP (S1 Table). Of these proteins, 73 had decreased and 52 increased expression in RER versus controls (0.05 to 1.60 $\log_2$ fold change (FC); P≤0.0133; FDR ≤0.05; Tables 2–4). Twenty-one of the proteins with the highest $\log_2$ FC involved blood-borne proteins such as hemoglobin and apolipoprotein A, suggesting increased hemorrhage in RER-susceptible muscle biopsies (S2 Table). Significant differential expression of proteins known to be a component of skeletal muscle ranged from -0.80 to 1.55 $\log_2$ FC. Of the 32 skeletal muscle proteins with increased differential expression, 11 involved calcium regulation and sarcoplasmic reticulum proteins (Table 2, Figs 1 and 2). This included carbonic anhydrase, present in both red blood cells and the sarcoplasmic reticulum, ankyrin 1 (ANK1) that connects sarcoplasmic reticulum membranes to the contractile apparatus and sarcolumenin and calsequestrin that bind $Ca^{2+}$ in the sarcoplasmic reticulum. RYR1, and calmodulin (CALM2), which regulates RYR1 also had increased expression. The two next largest protein classifications with increased expression each contained 4 proteins involved in ubiquitination/proteasomal degradation of proteins or antioxidants (Table 2, Figs 1 and 2). A lesser number of identified DEP were involved in glyco(geno)lysis, the plasma membrane, heat shock proteins and other functions (Table 2).

Of the 73 DEP with decreased expression, 47 were mitochondrial proteins many of which were subunits located in complexes I, II, III, IV and V of the electron transfer system (Table 3, Fig 3). The next largest class of muscle proteins with decreased DEP were contractile proteins, including the $Ca^{2+}$ binding troponins (TNN I, C, T) and calmodulin-regulated myosin regulatory light chain (MYL2) (Table 4, Fig 1). Z-disc and thin filament components, slow-twitch myosin light chains and fast-twitch type 2X myosin (MYH1) also had decreased DE (Table 4). There was no significant difference in slow twitch myosin (MYH7) or type 2A fiber myosin (MYH2) between RER-susceptible and control horses. Four heat shock proteins, 6 metabolic proteins, a slow twitch isoform of SERCA and ANXA2, a $Ca^{2+}$ binding protein involved in skeletal muscle membrane repair, also had decreased DEP (Table 4, Fig 2).

**GO Pathway analysis of differentially expressed proteins.** *Cellular component.* There were 27 significantly up-regulated GO terms in cellular components (S3 Table). Twenty-one of these GO terms contained 6–32 proteins involved in cellular membranes, with 20 of the GO terms including RYR1 (S3 Table). Other upregulated GO terms involved vesicles, extracellular matrix or blood microparticles (S3 Table). Nineteen GO terms were down-regulated relative to background protein expression and 18 involved mitochondrial processes that contained 8–24 proteins (S3 Table). One GO term involved the myelin sheath (S3 Table).

*Biological process.* There were 58 significant GO terms for biological processes that were upregulated relative to background expression in RER-susceptible versus control horses (S3 Table, Figs 4 and 5). Twenty-nine GO terms involved ion homeostasis or ion regulation of which 25 GO terms containing 6–32 proteins included RYR1 (S3 Table). Eight up-regulated GO terms containing 8–23 proteins involved protein metabolic processes/post-translational modification (S3 Table). The remaining up-regulated GO terms involved general biological processes and responses (S3 Table). Thirty GO terms were down-regulated relative to background expression (S3 Table). Fifteen GO terms containing 10–24 proteins involved mitochondrial energy metabolism and eight GO terms containing 32–35 proteins involved nucleotide or ribonucleotide processing (S3 Table, Figs 4 and 5). Seven GO terms involved general metabolic processes (S3 Table).

**Table 2. Differentially expressed muscle proteins (DEP) with increased expression comparing horses susceptible to recurrent exertional rhabdomyolysis with control.**

| Gene ID | Protein name | DEP | DEP |
|---|---|---|---|
| | | Log$_2$ FC | P-Value |
| **Calcium regulation and sarcoplasmic reticulum** | | | |
| CA1 | carbonic anhydrase 1 | 1.55 | 1.00E-04 |
| CA2 | carbonic anhydrase 2 | 1.10 | 1.00E-04 |
| ANK1 | ankyrin-1 isoform X4 | 0.24 | 1.70E-04 |
| CALM2 | calmodulin | 0.20 | 1.00E-04 |
| HRC | sarcoplasmic reticulum histidine-rich calcium-binding protein | 0.19 | 4.70E-03 |
| CAPN1 | calpain-1 catalytic subunit | 0.19 | 1.00E-04 |
| VCP | transitional endoplasmic reticulum ATPase | 0.13 | 1.00E-04 |
| SRL | sarcolumenin isoform X1 | 0.10 | 1.00E-04 |
| CASQ1 | calsequestrin-1 | 0.09 | 3.00E-03 |
| RYR1* | ryanodine receptor | 0.07 | 2.73E-02 |
| ANXA6 | annexin A6 isoform X1 | 0.05 | 9.00E-03 |
| **Protein alterations** | | | |
| UBE2V2 | ubiquitin-conjugating enzyme E2 variant 2 | 0.26 | 3.90E-04 |
| UBC | ubiquitin C | 0.26 | 6.50E-04 |
| PSMA4 | proteasome subunit alpha type-4 isoform X1 | 0.20 | 1.00E-03 |
| QDPR | dihydropteridine reductase isoform X1 | 0.18 | 9.50E-04 |
| GSN* | gelsolin | 0.15 | 1.00E-03 |
| PCMT1 | protein-L-isoaspartate (D-aspartate) O-methyltransferase | 0.1 | 9.00E-03 |
| **Antioxidants** | | | |
| PRDX2* | peroxiredoxin-2 isoform X1 | 1.08 | 1.00E-04 |
| SELENBP1 | selenium-binding protein 1 | 0.19 | 1.00E-04 |
| PRDX1 | peroxiredoxin-1 | 0.15 | 7.00E-03 |
| GLO1 | lactoylglutathione lyase | 0.10 | 3.00E-03 |
| **Plasma membrane** | | | |
| ATP1A2* | sodium/potassium-transporting ATPase subunit alpha-2 | 0.26 | 1.00E-03 |
| DMD* | dystrophin isoform X11 | 0.16 | 1.00E-03 |
| SNTB1 | beta-1-syntrophin isoform X2 | 0.11 | 2.00E-04 |
| **Glyco(geno)lysis** | | | |
| AGL* | Glycogen debrancher | 0.09 | 1.50E-04 |
| GAPDH | glyceraldehyde-3-phosphate dehydrogenase | 0.09 | 2.00E-03 |
| PHKA1 | phosphorylase b kinase regulatory subunit alpha | 0.05 | 5.50E-03 |
| **Heat shock proteins** | | | |
| ST13 | hsc70-interacting protein | 0.08 | 8.00E-03 |
| **Other** | | | |
| EIF4A2 | eukaryotic initiation factor 4A-II | 0.12 | 5.00E-03 |
| LOC106783470 | low quality protein | 0.80 | 2.00E-03 |
| LOC100147142 | low quality protein | 0.64 | 2.00E-03 |

Proteins are identified by their gene ID and protein name, FC indicates fold change. An asterisk indicates the encoding gene was also differentially expressed. Upregulated blood-born DEP are shown in S2 Table.

*Molecular function*. No molecular functions were upregulated. There were eight significant GO terms in molecular function containing 14–27 proteins that were downregulated in RER-susceptible versus control relative to background protein expression, all involved mitochondrial transporter activity (S3 Table).

**Table 3. Differentially expressed mitochondrial proteins (DEP) with decreased expression comparing horses susceptible to recurrent exertional rhabdomyolysis with control.**

| Gene ID | Gene Name | DEP log$_2$ FC | DEP P-Value |
|---------|-----------|----------------|-------------|
| **Mitochondrial** | | | |
| CKMT2 | creatine kinase S-type 2C | -0.23 | 1.00E-04 |
| SLC25A4 | ADP/ATP translocase 1 | -0.18 | 1.00E-04 |
| ACAT1 | acetyl-CoA acetyltransferase 2C | -0.18 | 1.00E-04 |
| CHCHD3 | MICOS complex subunit MIC19 isoform X2 | -0.15 | 3.00E-03 |
| PDHB | pyruvate dehydrogenase E1 component subunit beta 2C | -0.14 | 2.00E-03 |
| VDAC3 | voltage-dependent anion-selective channel protein 3 isoform X1 | -0.14 | 4.00E-03 |
| VDAC2 | voltage-dependent anion-selective channel protein 2 | -0.12 | 2.00E-03 |
| MDH2 | malate dehydrogenase 2C | -0.13 | 5.60E-04 |
| HADHA | trifunctional enzyme subunit alpha 2C | -0.13 | 1.00E-04 |
| CS | citrate synthase 2C | -0.13 | 6.90E-04 |
| CKM | creatine kinase M-type | -0.11 | 1.00E-04 |
| SLC25A12 | calcium-binding mitochondrial carrier protein Aralar1 isoform X1 | -0.11 | 1.00E-04 |
| VDAC1 | voltage-dependent anion-selective channel protein 1 | -0.10 | 1.10E-04 |
| NNT | NAD(P) transhydrogenase 2C mitochondrial isoform X1 | -0.09 | 1.00E-04 |
| ACADVL | very long-chain specific acyl-CoA dehydrogenase 2C soform X6 | -0.09 | 3.20E-04 |
| TUFM | elongation factor Tu 2C | -0.08 | 2.00E-03 |
| ACO2 | aconitate hydratase 2C | -0.07 | 1.00E-03 |
| PDHA1 | pyruvate dehydrogenase E1 component subunit alpha 2C somatic form 2C | -0.06 | 1.10E-02 |
| PHB2 | prohibitin-2 | -0.05 | 3.00E-03 |
| **Complex I NADH dehydrogenase [ubiquinone]** | | | |
| NDUFB11* | 1 beta subcomplex subunit 11 2C | -0.34 | 3.70E-04 |
| NDUFA3* | 1 alpha subcomplex subunit 3 | -0.19 | 6.00E-03 |
| NDUFS8* | iron-sulfur protein 8 2C | -0.19 | 1.00E-02 |
| NDUFA2 | 1 alpha subcomplex subunit 2 | -0.17 | 3.00E-03 |
| NDUFB5* | 1 beta subcomplex subunit 5 2C | -0.17 | 2.00E-03 |
| NDUFS3* | iron-sulfur protein 3 2C isoform X1 | -0.15 | 3.20E-04 |
| NDUFA9 | 1 alpha subcomplex subunit 9 2C | -0.80 | 1.20E-02 |
| NDUFA9 | 1 alpha subcomplex subunit 9 2C | -0.08 | 1.20E-02 |
| NDUFA10 | 1 alpha subcomplex subunit 102C isoform X1 | -0.10 | 1.00E-03 |
| NDUFA13 | 1 alpha subcomplex subunit 13 | -0.11 | 9.00E-03 |
| NDUFS1 | NADH-ubiquinone oxidoreductase 75 kDa subunit 2C isoform X1 | -0.10 | 1.50E-04 |
| **Complex II** | | | |
| SUCLA2 | succinate—CoA ligase [ADP-forming] subunit beta 2C | -0.13 | 1.00E-03 |
| SUCLG1 | succinate—CoA ligase [ADP/GDP-forming] subunit alpha 2C isoform X1 | -0.13 | 5.00E-03 |
| **Complex III** | | | |
| UQCRB | cytochrome b-c1 complex subunit 7 | -0.25 | 1.00E-04 |
| LOC111767815 | cytochrome b-c1 complex subunit 8 | -0.16 | 5.00E-03 |
| **Complex IV** | | | |
| COX6B1* | cytochrome c oxidase subunit 6B1 | -0.22 | 2.00E-03 |
| COX5B* | cytochrome c oxidase subunit 5B 2C | -0.17 | 1.00E-03 |
| COX5A* | cytochrome c oxidase subunit 5A 2C | -0.16 | 1.20E-04 |
| COX4I1 | cytochrome c oxidase subunit 4 isoform 12C | -0.09 | 1.00E-03 |
| **Complex V** | | | |
| ATP5IF1* | ATPase inhibitor 2Cisoform X1 | -0.38 | 1.00E-04 |
| ATP5PF | ATP synthase-coupling factor 62C | -0.27 | 8.10E-04 |
| ATP8 | ATP synthase F0 subunit 8 | -0.27 | 1.00E-03 |
| ATP5PD | ATP synthase subunit d 2Cisoform X1 | -0.24 | 1.00E-04 |

(*Continued*)

**Table 3.** (Continued)

| Gene ID | Gene Name | DEP log$_2$ FC | DEP P-Value |
|---------|-----------|----------------|-------------|
| ATP5PB | ATP synthase F(0) complex subunit B1 2C | -0.16 | 1.00E-04 |
| ATP5ME | ATP synthase subunit e 2C | -0.15 | 2.00E-03 |
| ATP5F1A | ATP synthase subunit alpha 2C | -0.14 | 1.00E-04 |
| ATP5F1C | ATP synthase subunit gamma 2C isoform X3 | -0.12 | 4.00E-03 |
| ATP5F1B | ATP synthase subunit beta 2C | -0.08 | 1.00E-03 |

Proteins are identified by their gene ID and protein name, FC indicates fold change. An asterisk indicates the encoding gene was also differentially expressed.

*KEGG pathways*. The down-regulated DEP was enriched for eight KEGG pathways. Seven of the pathways were related to mitochondrial proteins (i.e. oxidative phosphorylation, TCA-cycle, metabolic pathways, thermogenesis and Huntington, Parkinson and Alzheimer disease) and one was associated with muscle contraction (S3 Table, Fig 5). There were no down-regulated pathways enriched for KEGG.

**Table 4. Differentially expressed nonmitochondrial proteins (DEP) with decreased expression comparing horses susceptible to recurrent exertional rhabdomyolysis with control.**

| Gene Symbol | Gene Name | DEP log$_2$ FC | DEP P-Value |
|-------------|-----------|----------------|-------------|
| MYL3 | myosin light chain 3 | -0.49 | 1.00E-04 |
| MYL2 | myosin regulatory light chain 2%2C ventricular/cardiac | -0.48 | 1.00E-04 |
| TNNC1 | troponin C 2C slow skeletal and cardiac muscles | -0.39 | 1.00E-04 |
| TNNI1 | troponin I 2C slow skeletal muscle | -0.39 | 1.00E-04 |
| MYH1 | myosin-1 isoform X1 | -0.29 | 1.00E-04 |
| TNNT1 | troponin T 2C slow skeletal muscle isoform X1 | -0.26 | 1.00E-04 |
| MYBPC1 | myosin-binding protein C 2C slow-type isoform X2 | -0.15 | 1.00E-04 |
| MYOZ1 | myozenin-1 | -0.14 | 1.00E-03 |
| CFL2 | cofilin-2 isoform X1 | -0.14 | 1.00E-03 |
| MYOT | myotilin isoform X1 | -0.09 | 1.00E-03 |
| FLNC | filamin-C isoform X1 | -0.05 | 1.00E-03 |
| ACTN3 | alpha-actinin-3 | -0.10 | 1.00E-04 |
| LOC100070523 | 10 kDa heat shock protein 2C mitochondrial isoform X2 | -0.25 | 1.00E-04 |
| HSPB1 | heat shock protein beta-1 | -0.19 | 1.00E-04 |
| CRYAB | alpha-crystallin B chain | -0.19 | 1.20E-04 |
| HSPA9 | stress-70 protein 2C mitochondrial | -0.08 | 1.00E-03 |
| MDH1 | malate dehydrogenase 2C cytoplasmic | -0.17 | 1.00E-04 |
| FBP2 | fructose-1 2C6-bisphosphatase isozyme 2 | -0.11 | 3.00E-03 |
| TPI1 | triosephosphate isomerase | -0.08 | 5.90E-03 |
| ALDH1A1 | retinal dehydrogenase 1 | -0.08 | 1.00E-04 |
| CKM | Creatine kinase-M | -0.11 | 1.00E-04 |
| ENO3 | beta-enolase | -0.07 | 1.10E-02 |
| ATP2A2 | sarcoplasmic/endoplasmic reticulum calcium ATPase 2 isoform X1 | -0.27 | 1.00E-04 |
| ANXA2 | annexin A2 isoform X1 | -0.24 | 3.40E-04 |
| MB | myoglobin | -0.31 | 1.00E-04 |
| KPNA3 | importin subunit alpha-4 isoform X1 | -0.28 | 3.00E-03 |

Proteins are identified by their gene ID and protein name, FC indicated fold change. The log$_2$ fold change and *P* value are shown for the protein. An asterisk indicates that the encoding gene was also differentially expressed.

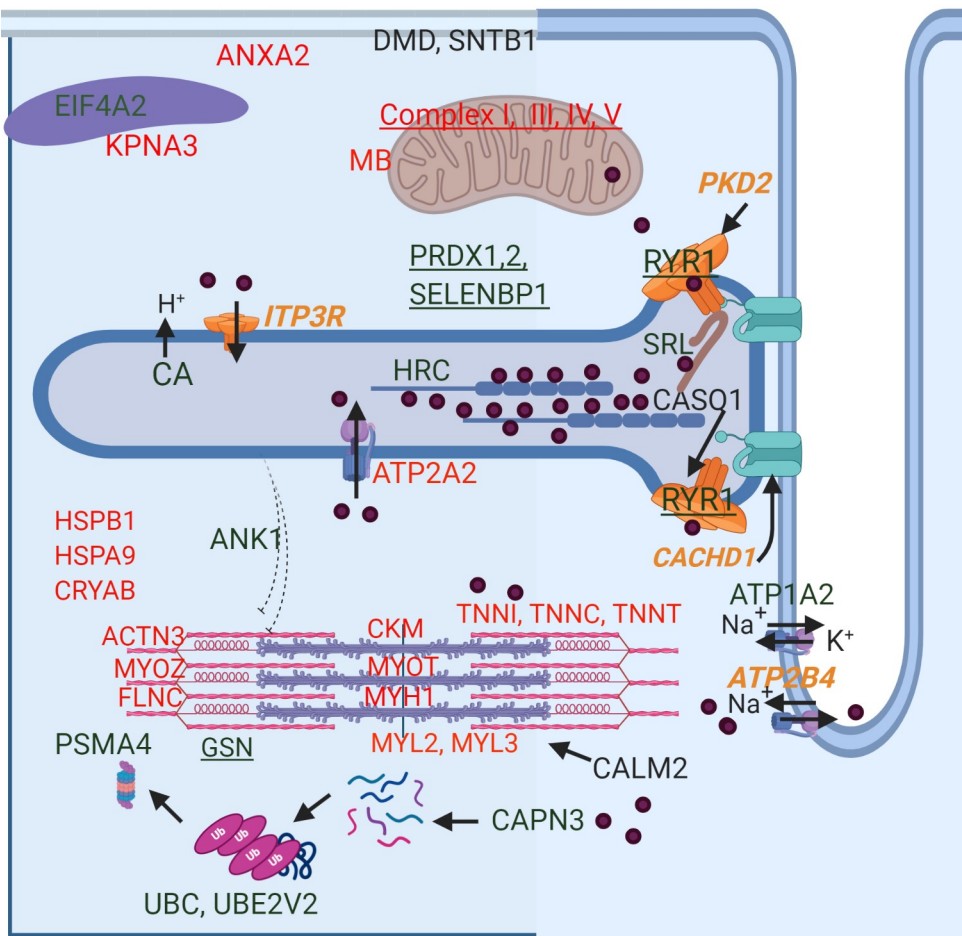

**Fig 1. Depiction of the cellular location of differentially expressed proteins and select differentially expressed genes in RER-susceptible horses.** Increased expression is in green and decreased expression in red. Proteins that also had differential expression of their encoding gene are underlined. Differentially expressed genes (*italics*) relating to $Ca^2$ $^+$regulation are shown in orange. $Ca^{2+}$ is depicted by purple circles. Full names of genes and proteins are found in Tables 2–4. RER-susceptible horses between episodes of rhabdomyolysis had increased expression of ANK1, which links sarcoplasmic reticulum to myofilaments, $Ca^{2+}$ binding proteins within the sarcoplasmic reticulum (CASQ, SRL, HRC), the $Ca^{2+}$ release channels *ITP3R* and RYR1, as well as regulators of RYR1 [calmodulin (CALM2), calsequestrin (CASQ1), *PKD2*, and *CACHD1* (via DHPR)]. Proteins involved in oxidative stress (PRDX1, PRDX2, SELENBP1) and protein degradation [$Ca^{2+}$-activated calpain (CAPN3), ubiquitination (UBC, UBE2V2), proteasome (PSMA4)] were differentially expressed. Decreased expression was found for $Ca^{2+}$ activated proteins [SERCA2 (*ATP2A2*), troponins (TNN1,C,T) myosin regulatory light chains (MYL2,3)] as well as other sarcomere proteins (ACTN3, MYOZ, FLNC, MYH1). Heat shock proteins (HSPB1, HSPA9) and protein chaperones had decreased expression in RER. Decreased expression was found for mitochondrial proteins myoglobin (MB), 6 subunits of complex I, 3 subunits of complex III, 4 subunits of complex IV and 4 subunits of complex V. ANXA2, a $Ca^{2+}$ activated membrane repair protein was also downregulated and sarcolemmal structural proteins dystrophin (DMD) and syntrophin (SNTB1) were upregulated. The sodium/potassium exchanger (ATP1A2) and the sodium/ $Ca^{2+}$ exchanger *ATP2B4*, were also differentially expressed.

## Transcriptomics

**Expressed genes.** *RER*. A total of 28,625 full-length transfrags were quantified from ~45 million short-read pairs per library. Adapter and quality filtering removed 27.4% of reads. Of the ~33 million read pairs retained after quality filtering, 74.5% mapped to the reference genome EquCab3.0. Only the uniquely mapped reads were used to quantify transcript

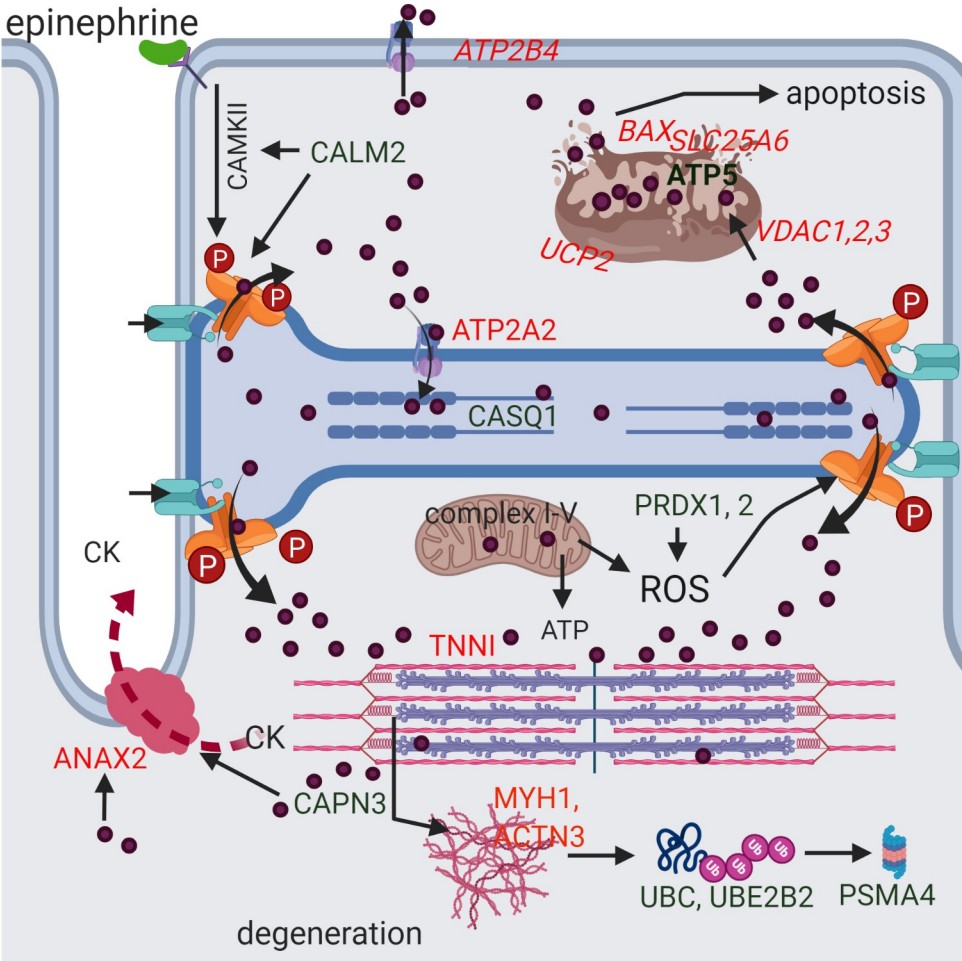

**Fig 2. A potential scenario for the development of rhabdomyolysis in RER-susceptible horses derived from differentially expressed proteins and genes.** Green indicates increased and red decreased differential expression. Genes are italicized. A stressful environment is proposed to induce hyper-phosphorylation of RYR1 through beta adrenergic (epinephrine mediated) activation of calmodulin kinase II (CAMKII) which is stimulated by the DEP calmodulin (CALM2). This and other potential post-translational modifications of RYR1 (oxidation/nitrosylation) allow excessive $Ca^{2+}$ (purple circles) release of high sarcoplasmic reticulum $Ca^{2+}$ stores in RER-susceptible horses through RYR1 (orange) when RYR1 opening is stimulated during exercise by the voltage gaited $Ca^{2+}$ channel in the t-tubule (turquoise). Myoplasmic $Ca^{2+}$ interacts with troponin I (TNNI) and, when not adequately pumped back into the sarcoplasmic reticulum (ATP2A2) or out of the cell (*ATP2B4*), $Ca^{2+}$ produces persistent contracture of the sarcomere (pink and purple). Mitochondria buffer myoplasmic $Ca^{2+}$ through *VDAC* uptake where $Ca^{2+}$ stimulates ATP production. However, in excess, $Ca^{2+}$ uncouples (*UCP2*) oxygen consumption from electron transport and ATP production and releases reactive oxygen species (ROS). Antioxidants such as peroxiredoxin (PRDX1, PRDX2) counteract ROS to prevent oxidative stress. Excessive mitochondrial matrix $Ca^{2+}$ results in formation of the membrane transition pore (*BAX*, *SLC25A6*, ATP5), loss of membrane potential, release of cytochrome C and apoptosis. Calcium activation of proteases such as calpain (CAPN3) results in degradation of myofilaments, cellular proteins and cell membranes resulting in the release of muscle proteins such as creatine kinase (CK) into the blood stream. Calcium activated ANAX2 participates in repair of membranes that have increased dystrophin (DMD) and syntropin (SNTB1) content. Degraded proteins are ubiquitinated (UBC, UBE2B2) and processed in the proteasome (PSMA4).

abundance (~24 million read pairs, 53% of total sequenced read pairs). The average depth of coverage per sequenced base was 60.0X ± 7.8. Of the 28,623 expressed transcripts, 96.5% were determined to be autosomal and 3.5% were mapped to the X chromosome. After filtering for low count transcripts, 14,155 (49.4%) remained for the principal component and differential expression analysis.

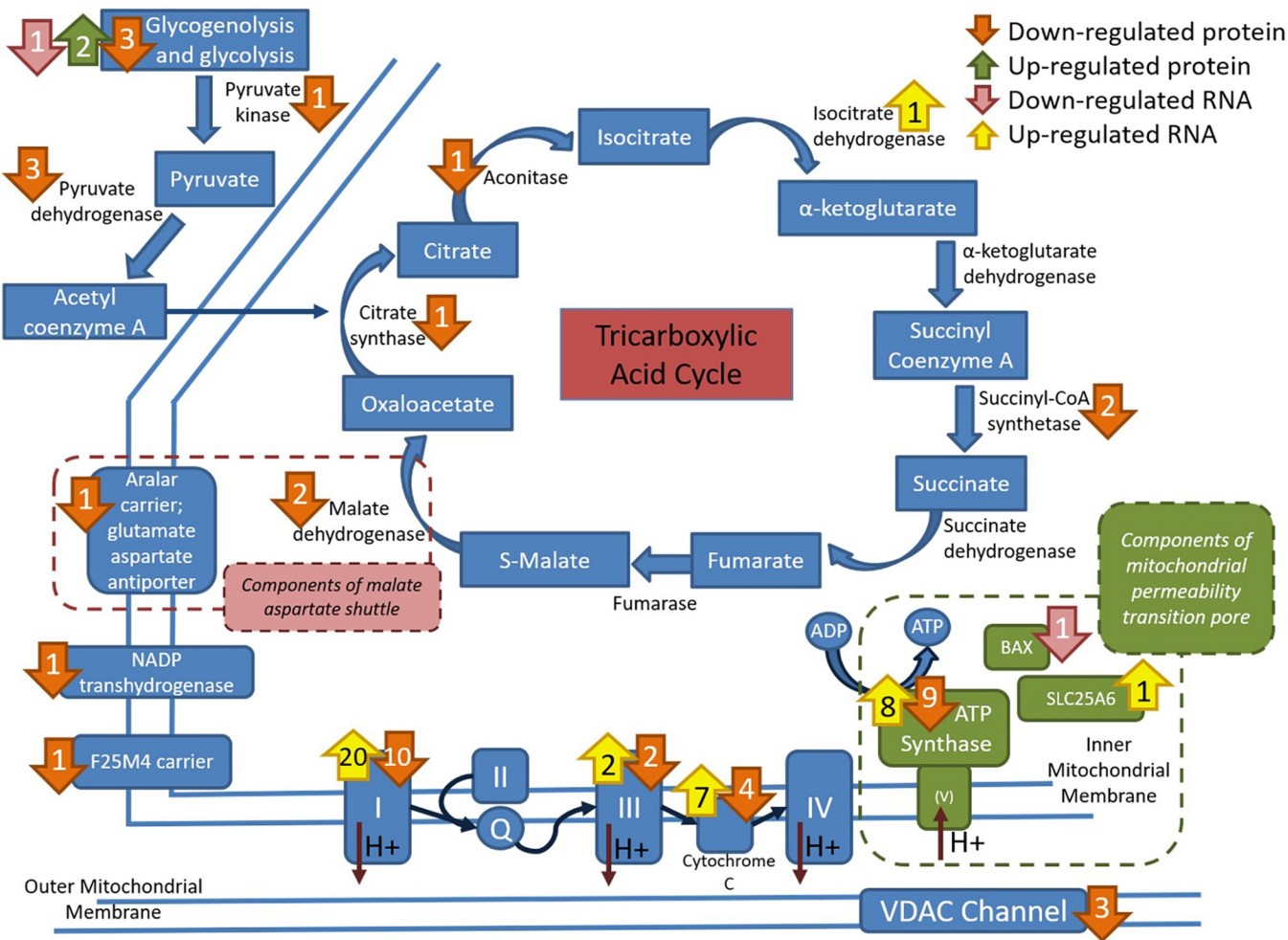

**Fig 3. Depiction of the location of differentially expressed proteins and genes within the mitochondrial tricarboxylic acid cycle, electron transport system and mitochondrial Ca²⁺regulation.** Increased expression is in green for protein and yellow for genes. Decreased expression is in orange for proteins and pink for genes. Note the strong impact of RER-susceptibility on the electron transfer system and channels related to Ca²⁺ uptake (*VDAC*) or overload (mitochondrial permeability transition pore).

*Principal component analysis of DEG.* The RER-D and control group clustered together in the principal components (PC) 1 and 2 and were separate from the RER-susceptible group which showed larger within-group variability. The first three PC accounted for 19.1, 15.5 and 10.0% of the gene expression variance, respectively (Fig 5). The separation between RER-susceptible and control was lost in the PC1 and 3 comparison, where all phenotypes clustered together (Fig 5). These results show that the largest variability in gene expression corelates to differences between RER-susceptible and control horses.

*RER-D.* There were no differentially expressed transcripts comparing RER-D to controls, as expected from the PCA analysis (Fig 6).

*RER-susceptible differential gene expression.* There were a total of 812 DEG; 466 were up-regulated and 346 were down-regulated in RER-susceptible versus controls (range -3.23 to 3.42 log₂ FC, FDR ≤ 0.05) (S4 Table). Ninety-eight transcripts (56⁻and 42 - DEG) were annotated to locus identifiers and 32 were novel transcripts (22⁻and 10 - DEG) unannotated in the current reference genome. The two annotated transcripts that had > 2 log₂ FC -DEG were

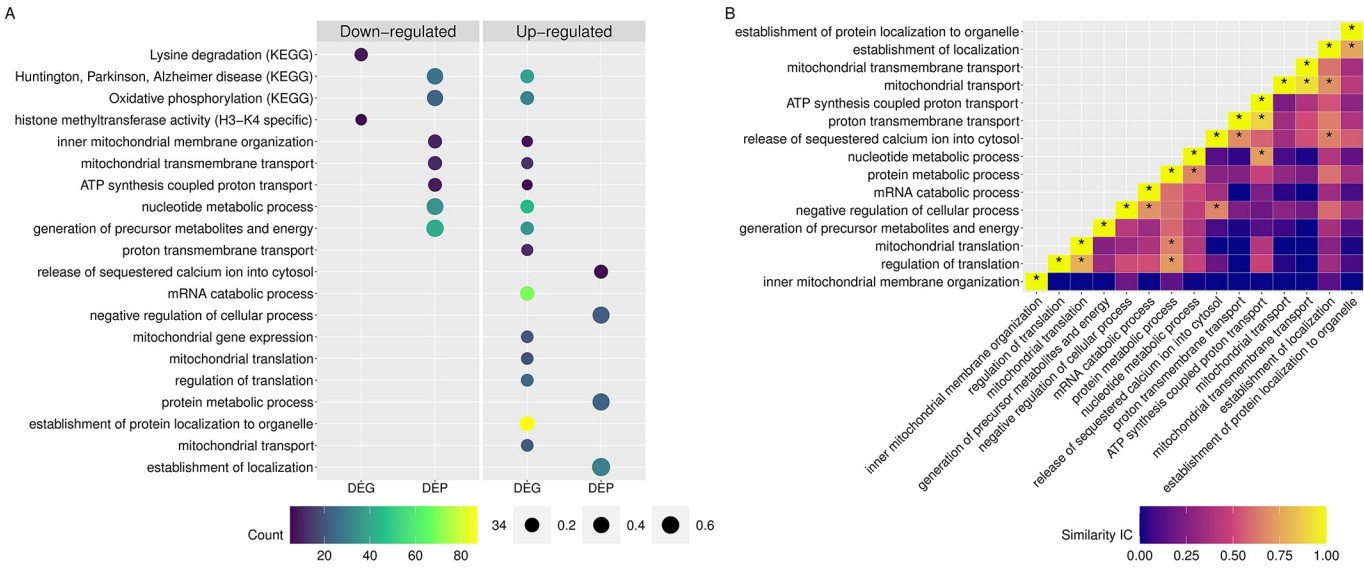

**Fig 4.** (A) GO and KEGG pathways enriched for DEG and DEP. The size of the dots represents the ratio of DEG or DEP to the background used in the enrichment analysis. Since the background of DEG differs from that used for DEP, the points are color coded to note the number of DEG or DEP in each pathway. (B) Heatmap of the information content (IC) shared between two pathways. The asterisk (*) denotes two pathways that share an IC > 0.65.

*TTLL11* (log$_2$ FC 2.09), a polyglutamase that preferentially modifies alpha-tubulin and *OLFM1* (log$_2$ FC 2.07), an olfactory gene. The two genes that had > 2 log$_2$ FC DEG were *BAX* (log$_2$ FC -2.82), a component of the mitochondrial transition pore and an apoptotic activator and *MYO7A* (log$_2$ FC -2.53), active in intracellular movement/trafficking. With a large number of

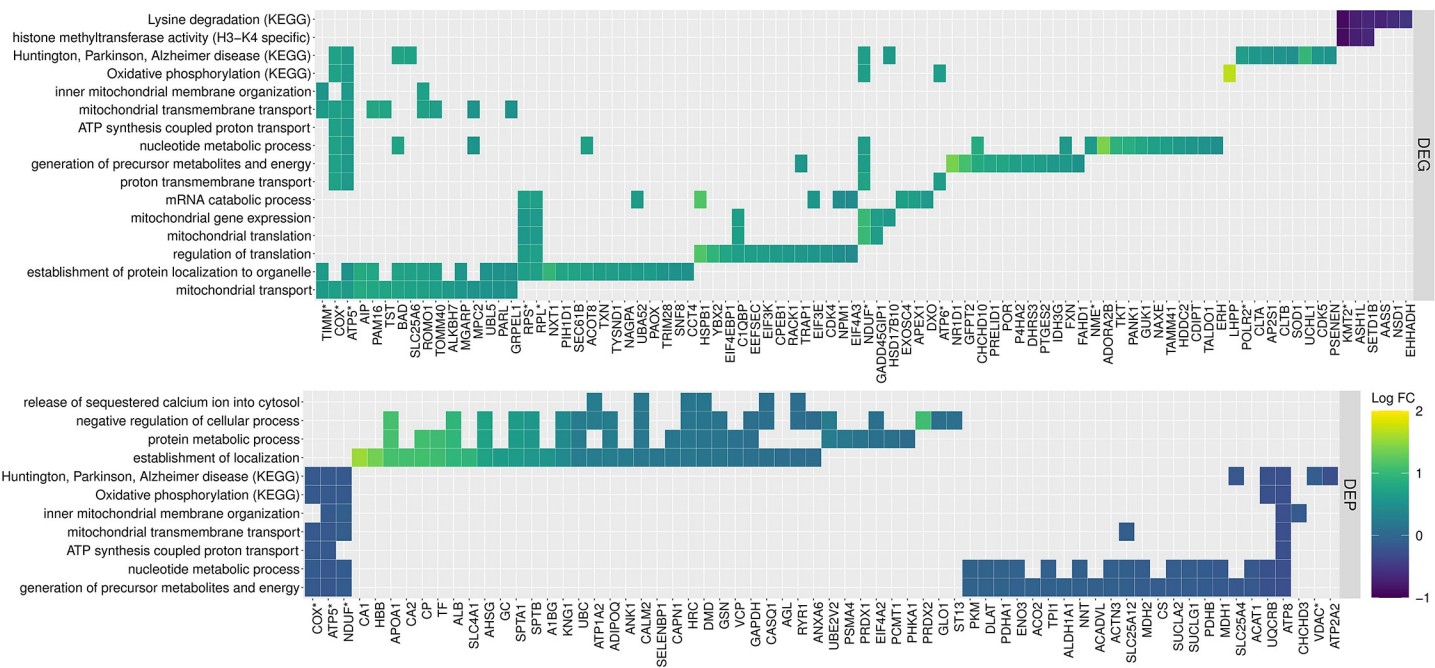

**Fig 5. Heatmap illustrating the individual DEG and DEP found within enriched GO and KEGG pathways.** Darker colors indicate lower log$_2$ fold change difference in gene or protein expression in RER-susceptible compared to control horses.

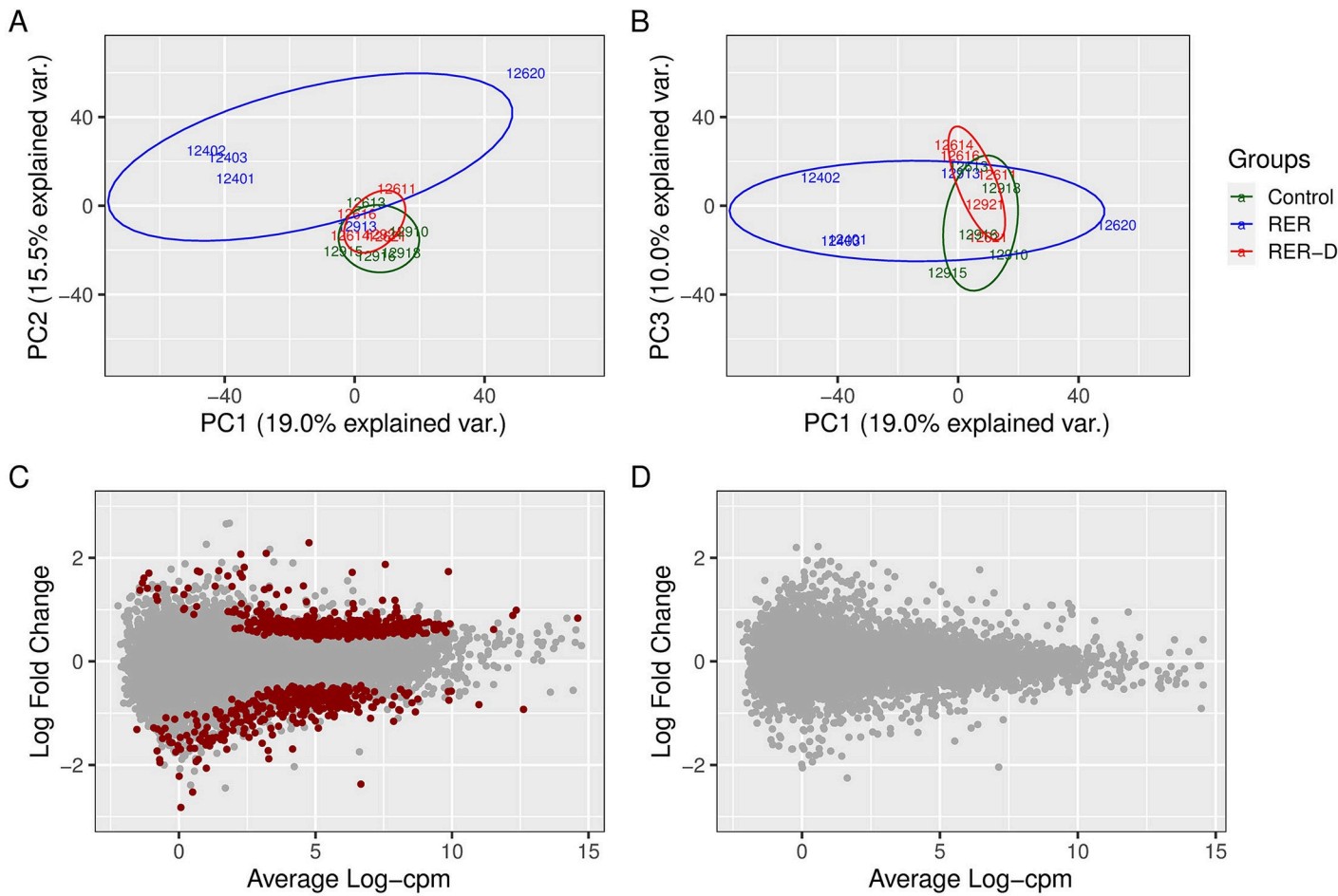

**Fig 6. Principal component (PC) analysis of transcriptomic data for control, RER-susceptible (RER) and RER horses treated with dantrolene (RER-D).** (A) PC1 versus PC2. Note that RER-susceptible horses cluster separately from overlapping RER-D and controls. (B) PC1 and PC3. The proportion of variance explained by the RER and control groups is reduced by PC3. (C) MA plot depicting the average expression of gene transcripts compared to the $\log_2$ fold change of differentially expressed genes comparing RER-susceptible to controls. There were 812 differentially expressed genes (red dots) between RER-susceptible and control horses. (D) MA plot showing the average expression of gene transcripts compared to the $\log_2$ fold difference in differentially expressed genes in RER-D compared to controls. There were no significant differentially expressed genes observed between RER-D and controls.

DEG in our dataset having modest $\log_2$ FC, we focused our assessment on the functional classification of DEG using GO analyses.

**Gene Ontology (GO) pathway for RER transcriptomic analysis.** *Cellular component.* There were 48 up-regulated enriched GO terms identified in cellular components for RER-susceptible horses (S5 Table). Twenty-one GO terms containing 5–65 genes, involved the mitochondrion, primarily the electron transfer system, 17 GO terms containing 6–84 genes involved the ribosome and the remainder involved a variety of functions from rough endoplasmic reticulum to adhesion (S5 Table). In total, 3% of DEG were ribosomal. No cellular components were down-regulated.

*Biological process.* There were 104 upregulated significant GO terms identified in biological processes for RER-susceptible versus control horses relative to skeletal muscle background gene expression (S5 Table). Mitochondrial processes had the largest number of GO terms at 50 terms containing 5–40 genes (S5 Table, Figs 4 and 5). Other significant GO terms involved catabolic processes (13 GO terms, 4–74 genes) protein targeting to an organelle/ribosome (25 GO

terms, 4–87 genes), translation (nine GO terms, 11–68 genes) and others (S5 Table, and Figs 4 and 5). No biological processes were down-regulated.

*Molecular function*. There were 22 significant GO terms upregulated in molecular function in RER-susceptible versus control relative to background expression (S5 Table). The up-regulated pathways primarily involved the mitochondrial electron transfer system (14 GO terms, 4–41 genes), with others involving the ribosome (4 GO terms), mRNA binding (2 GO terms), and translational regulation (1 GO term) (S5 Table). There was one down-regulated GO term, histone methyl transferase (H3-K4 specific) which incorporated 5 genes (*KMT2A*, *KMT2B*, *KMT2D*, *ASH1*, *SETD1B*) (S5 Table).

*KEGG pathways*. Eight KEGG pathways were enriched among the up-regulated DEG. The top five pathways were ribosome, oxidative phosphorylation, and disease pathways associated with mitochondrial dysfunction (i.e. Huntington, Parkinson and Alzheimer disease). The remaining enriched pathways were related to thermogenesis, non-alcoholic fatty liver disease and retrograde endocannabinoid signaling. The only pathway enriched for down-regulated DEG was lysine degradation (Figs 4 and 5). This pathway is related to the H3-K4 specific histone methyl transferase seen enriched in molecular function (Figs 4 and 5).

## Differentially expressed genes with differentially expressed proteins

Nineteen DEG encoded DEP (Figs 1 and 2). This included *RYR1*, heat shock protein *HSPB1*, glycogen debrancher *AGL*, plasma membrane dystrophin *DMD*, cytoskeleton element spectrin S*PTB*, and the Na/K exchanger *ATP1A2* preferentially concentrated in T-tubular membranes, all with ↓DEG and ↑DEP. Six subunits of complex I, one subunit of complex III, 3 subunits of cytochrome C oxidase and one inhibitor of complex V had ↑DEG and ↓DEP (Fig 3). The antioxidant peroxiredoxin 2 (PRDX2) and gelsolin (GSN), an actin-modulating component, were up-regulated as genes and proteins.

## Semantic similarities between enriched biological processes

Biological processes enriched for either or both DEG and DEP were further evaluated to identify shared functionality associated with RER susceptibility (Figs 4 and 5). There were 19 enriched terms up-regulated for DEG and down-regulated for DEP, encompassing 90 DEG (log$_2$ FC ranged from 0.5 to 1.7) and 49 DEP (-0.05 to -0.38 log$_2$ FC) (Fig 4A). The pathways with antagonistic expression patterns at a single timepoint related to mitochondrial processes including oxidative phosphorylation, mitochondrial organization and transmembrane transport (Fig 4, S3 and S5 Tables). The top GO terms with a similarity Conrath's Information Content greater than 0.65 were upregulated in both the transcriptome and proteome (Fig 4B). The genes and proteins associated with these pathways included subunits of complex I NADH ubiquinone oxidoreductase (20 -DEG and 10 DEP), complex IV cytochrome c oxidase (4 -DEG and 4 DEP), complex V ATP synthase (8 -DEG and 10 DEP), mitochondrial transmembrane channel DEG TIM and TOM translocases, located in the inner (*TIMM10*, *PAM16*, *TIM17B*, and *TIMM50*) and outer mitochondrial membrane (*TOMM40*), and the DEP VDAC1 and VDAC2, voltage-dependent anion channel (Fig 5). Additional genes of importance associated with the pathways exhibiting antagonistic expression, include genes related to mitochondrial protein synthesis like the DEGs *HSD17B10* (mitochondrial ribonuclease P), required for tRNA maturation [35] and *PAM16* and *TST* (required for the import of nuclear-encoded mitochondrial mRNA [36] and rRNA [37] respectively). DEG associated with proton transmembrane transport and mitochondrial transport were functionally similar to DEP enriched for the release of sequestered Ca$^{2+}$ into the cytosol and establishment of localization. The negative regulation of cellular processes enriched for DEP was associated with mRNA catabolic processes

enriched for DEG. Similarly, DEG enriched for mitochondrial translation and the regulation of translation were related to DEP associated with protein metabolic processes (Fig 5). Taken together these results highlight the interdependence of nuclear gene expression and mitochondrial protein translation in the regulation of energy metabolism when mitochondrial integrity may be compromised.

## Discussion

Proteomic and transcriptomic comparisons of control and RER-susceptible mares housed in the same race training facility indicated that RER-susceptible horses have altered expression of proteins and genes predominantly involved in skeletal muscle $Ca^{2+}$ regulation/$Ca^{2+}$ binding, the sarcomere and the mitochondrion. Flux of $Ca^{2+}$ between the SR and the myoplasm is essential for muscle contraction and for mitochondrial $Ca^{2+}$ signaling to stimulate ATP production [38]. Our results suggest that regulation of myoplasmic $Ca^{2+}$ homeostasis is perturbed in horses susceptible to- but not currently experiencing RER- when housed in a race-training environment. Marked alterations in the expression of mitochondrial proteins in RER susceptible horses could be related to the role of mitochondria in buffering myoplasmic $Ca^{2+}$ which could have a detrimental down-stream effect on the mitochondrial respiratory chain and apoptosis [39].

Under resting conditions, myoplasmic $Ca^{2+}$ concentrations are maintained at nanomolar levels by pumping $Ca^{2+}$ out of the myofiber or into the sarcoplasmic reticulum where it is bound to luminal proteins until an action potential stimulates $Ca^{2+}$ release [38,40,41]. Alterations in expression of the $Ca^{2+}$ pump (ATP2A2), the plasma membrane $Ca^{2+}$ transporting ATPase (*ATP2B4*), high affinity sarcoplasmic reticulum $Ca^{2+}$ binding proteins [calsequestrin (CASQ1), sarcolumenin (SRL), sarcoplasmic reticulum histidine-rich $Ca^{2+}$ binding protein (HRC)] in RER-susceptible horses suggests that RER muscle has a decreased capacity to remove $Ca^{2+}$ from the myoplasm and enhanced capacity to sequester and release sarcoplasmic reticulum $Ca^{2+}$ [42]. Modest increases in myoplasmic $Ca^{2+}$ release during muscle contraction could increase actin-myosin interaction and contractile force which could contribute to the superior performance indices reported for RER-susceptible Swedish trotters [43].

In horses experiencing rhabdomyolysis, myoplasmic $Ca^{2+}$ concentrations 8-fold higher than normal have been reported [14]. Excessive release of myoplasmic $Ca^{2+}$ through sustained opening of RYR1, similar to that seen with malignant hyperthermia, is a common cause of elevated myoplasmic $Ca^{2+}$ and rhabdomyolysis [44]. RYR1 was an ↑ DEP and both sarcoplasmic reticulum $Ca^{2+}$ release channels *RYR1* and inositol 1,4,5-triphosphate receptor (*ITPR3*) were DEG in RER-susceptible horses, with ↑ *RYR1* DEG also found in a previous study of Standardbred trotters with RER. Other key regulators of RYR1, including calmodulin (CALM2) [45], calsequestrin (CASQ1) [46], polycystin (*PKD2*) [47] and cache domain containing 1 (*CACHD1)* acting via the dihydropyridine receptor were also all DEP or DEG [38]. Thus, proteomic and transcriptomic analyses support a role for RYR1 and interacting proteins in RER susceptibility (Figs 1 and 2). Several other $Ca^{2+}$-sensitive proteins had altered expression in RER-susceptible versus control muscle including calpain (CAPN3), a $Ca^{2+}$ activated protease and carbonic anhydrase (CA), which is believed to enhance the kinetics of $Ca^{2+}$ transport through its effect on sarcoplasmic reticulum counter transport of $H^+$ (Fig 1) [48]. Horses experiencing exertional rhabdomyolysis have previously been shown to have 56 times higher serum carbonic anhydrase concentration than healthy Thoroughbreds, supporting a muscular origin for this enzyme [49].

Linkage analysis and genome-wide association studies have not found an association between RER and RYR1 [5,6,50]. The open probability of the RYR1 channel, however, is

impacted by diverse post-translational modifications (e.g., phosphorylation, oxidation and nitrosylation) with post-translational modification being an upregulated GO biologic process term in RER-susceptible horses [38,51,52]. The DEP calmodulin (CALM2) regulates phosphorylation of RYR1 via calmodulin kinase II (Fig 2) and phosphorylation of RYR1 occurs with beta-adrenergic stimulation and chronic stress, conditions linked to RER-susceptibility and racing [1,51]. Chronic stress in a race training environment has been demonstrated by cortisol concentrations which interestingly, similar to the incidence of RER, were higher in younger versus older mares in race training [53]. Conditions of chronic stress and associated prolonged beta-adrenergic stimulation could produce post-translational modification of RYR1 in young fillies that trigger enhanced $Ca^{2+}$ release in RER-susceptible horses. Further, post-translational modification of RYR1 could explain why a direct genetic link between RYR1 and RER disease expression has not been identified (Fig 2). Dantrolene, which binds to RYR1 and decreases $Ca^{2+}$ release, is often used to treat refractory cases of RER [13–15]. In our study, RER horses treated with dantrolene, showed no DEG compared to control horses further supporting a role for RYR1 and $Ca^{2+}$ dysregulation in RER.

Mitochondria located near RYR1 modulate myoplasmic $Ca^{2+}$ by serving as major sites of $Ca^{2+}$ uptake and buffering [54,55]. All three genes encoding isoforms of the voltage-dependent anion channels (*VDAC*) that allow $Ca^{2+}$ access to the mitochondrial intermembrane space were down-regulated in RER-susceptible horses (Figs 1–3). $Ca^{2+}$ entry into the mitochondrial matrix is impacted by VDAC expression [56] and occurs through the mitochondrial $Ca^{2+}$ uniporter complex (MCU) with the driving force being the steep mitochondrial inner membrane potential maintained by electron transfer system complexes I, III and IV [38]. These complexes had altered gene and protein expression in RER-susceptible horses (Fig 3). Thus, DEP and DEG support a role for $Ca^{2+}$ buffering by mitochondria in RER-susceptible horses.

When mitochondrial $Ca^{2+}$ exceeds physiologic levels, mitochondrial $Ca^{2+}$ overload occurs activating formation of the mitochondrial permeability transition pore complex [57]. Pore formation causes loss of inner membrane potential and eventual organelle swelling and rupture (Fig 2) [57]. All three proposed components of this pore were DEP or DEG in RER-susceptible horses including ATP synthase (*ATP5*, complex V), the adenine nucleotide translocator (*SLC25A6*, ANT), and proapoptotic Bcl-2 family member *BAX* [58]. Thus, proteomic and transcriptomic analyses support alteration in the electron transfer system and the mitochondrial permeability transition pore linked to mitochondrial $Ca^{2+}$ overload as components of RER-susceptibility (Figs 1 and 2).

One likely consequence of mitochondrial $Ca^{2+}$ overload in RER-susceptible horses is uncoupling of electron transport across the inner mitochondrial membrane [59]. The gene encoding uncoupling protein 4 (*SLC25A27*) and *VPS13C*, involved in maintaining mitochondrial membrane potential, had⁻DEG in RER-susceptible horses. In RER Standardbreds, decreased expression of the gene encoding uncoupling protein 2 (*UCP2*) [10] was identified and lower coupling of the electron transfer system was found in 2 RER-susceptible horses based on flux control [17]. While a degree of uncoupling of ATP-production may thus be present in RER, a persistent disruption of mitochondrial ATP generation is unlikely. Reasons for this include that RER-susceptible horses can be top performers, RER horses have normal activities of respiratory complex activity, and RER horses have normal lactate accumulation with near-maximal exercise [43,60,61]. Additionally, the magnitude of DEP was small across mitochondrial proteins with only a fraction of the total subunits comprising each complex impacted. Changes in subunits of mitochondrial complexes and uncoupling proteins could, however, have a subtle or intermittent impact on mitochondrial function.

Of the DEG identified in our study only 16 had encoded proteins that were also DEP. Notably, these DEP largely had opposite directions of differential expression. RER-susceptible

horses in particular appeared to have an induced transcriptional and translational response directed towards restoring mitochondrial protein loss and improving mitochondrial biogenesis. Several genes encoding ribosomal proteins in the cytoplasm (*RPS*, 27 DEG; *RPL*, 36 DEG) and mitochondria (*MRPS*, 8 DEG; *MRPL*, 8 DEG), as well as genes associated with mRNA processing and transport were up-regulated in RER-susceptible horses compared to controls. The translation initiation factor 3 complex (*EIF3E*, *EIF3G* and *EIF3K*), the negative RNA splicing factor exon junction complex (*EIF4A3*), the mitochondrial ribonuclease (Rnase) P complex (*HSD17B10*), and the mitochondrial translation elongation factor (*TUFM*), all had altered expression, further supporting increased signaling for mitochondrial protein translation [62]. Similarly, several genes involved in the transport of nuclear-encoded mitochondrial proteins (*PAM16 and* TIM/TOM translocases) and rRNA for protein synthesis (rhodanese, *TST*), were up-regulated in RER-susceptible horses. Notably, however, many mitochondrial proteins were down regulated in RER-susceptible horses suggesting either increased protein degradation (supported by increased proteasomal gene and protein expression) and/or translational repression. Increased expression of the translational repressors DEP EIF4A2 and DEG *EIF4EBP1* and DEG enriched for mRNA catabolic processes support increased translational repression of aberrant RNA in RER [63,64]. Further investigation is needed into the specific processes involved in down-regulating mitochondrial proteins in RER-susceptible horses.

Limitations of the present study included biopsies that contained muscle fibers, nerve branches, blood vessels and blood contents, which complicates discerning myocyte-specific responses. Many of the top DEP in RER-susceptible muscle biopsies were blood-borne proteins. It is possible that more blood-borne proteins were the results of greater blood flow post-exercise and hemorrhage in RER-susceptible horse muscle as a result of heightened angiogenesis following multiple episodes of myodegeneration. Another limitation of our study was the small number of horses and the fact that the effect of dantrolene could not be evaluated in the same RER horses due to individual veterinarian's discretion over treatment. The dose of dantrolene used in the present study is similar to the dose (800 mg/horse/day) previously been shown to lower serum CK activity following exercise in RER-susceptible Thoroughbreds [15]. Dantrolene is often used for the most difficult to control RER horses which was confirmed in the present study by the higher range of CK activity in RER-D horses. This makes it quite remarkable that there were no DEG in horses with difficult to control RER given dantrolene compared to control horses. Ideally, all horses would have been the same age and at the same stage of training, however, all horses were in full race training and there were no significant differences in ages between groups. Utilizing these well-characterized phenotypic extremes, however, with age, sex, training and environment controlled to the extent possible in a field trial, we were able to identify subtle yet important differences in gene and protein expression between control and RER-susceptible horses.

In conclusion, based on the differential expression patterns of specific genes/proteins and GO pathways in RER-susceptible horses, we propose a scenario whereby horses susceptible to RER have enhanced storage of $Ca^{2+}$ in the sarcoplasmic reticulum, which enhances muscle force and energy metabolism within a physiologic range of myoplasmic $Ca^{2+}$. However, under conditions of chronic stress, we hypothesize that exercise and beta-adrenergic stimulation could cause post-translational modification of RYR1 producing intermittent episodes of prolonged opening of RYR1, excessive release of $Ca^{2+}$ into the myoplasm, persistent muscle contracture, excessive mitochondrial $Ca^{2+}$ uptake, uncoupling of the mitochondrial electron transfer system and loss of mitochondrial and myofiber integrity (Fig 2). Increased mitochondrial transcriptional/translational responses coupled with decreased mitochondrial protein content in RER-susceptible horses could indicate increased protein degradation and compensatory upregulation of mitochondrial protein production.

## Supporting information

**S1 Table. All differentially expressed proteins (DEP) in RER-susceptible compared to control horses.** Information is provided for average protein spectra for RER-susceptible and controls groups, GenBank protein accession, molecular weight and permutation test results.
(XLSX)

**S2 Table. Differentially expressed blood-borne proteins (DEP) in RER-susceptible compared to control horses.**
(XLSX)

**S3 Table. Significant up and down-regulated Molecular, Cellular and Biological Gene Ontology and KEGG pathways for DEP in RER-susceptible compared to control horses.** Additional information on enriched GO terms and KEGG pathways including descriptions, gene and background ratios, test statistics and enriched genes are provided.
(XLSX)

**S4 Table. Differentially expressed gene transcripts (DEG) in RER-susceptible compared to control horses.** Information on the mapped position of DEG, the average gene transcript expression for RER-susceptible and controls groups, and additional test statistics is provided.
(XLSX)

**S5 Table. Significant up and down-regulated Molecular, Cellular and Biological Gene Ontology and KEGG pathways for DEG in RER-susceptible compared to control horses.** Additional information on enriched GO terms and KEGG pathways including, descriptions, gene and background ratios, test statistics and enriched genes are provided.
(XLSX)

## Acknowledgments

The contribution of Dr. James Slaughter and technical assistance of Keri Gardner is greatly appreciated.

## Author Contributions

**Conceptualization:** Clara Fenger, Stephanie J. Valberg.

**Data curation:** Deborah Velez-Irizarry.

**Formal analysis:** Kennedy Aldrich, Deborah Velez-Irizarry, Melissa Schott.

**Investigation:** Kennedy Aldrich, Clara Fenger, Stephanie J. Valberg.

**Methodology:** Deborah Velez-Irizarry, Melissa Schott.

**Project administration:** Stephanie J. Valberg.

**Resources:** Stephanie J. Valberg.

**Writing – original draft:** Kennedy Aldrich.

**Writing – review & editing:** Kennedy Aldrich, Deborah Velez-Irizarry, Clara Fenger, Melissa Schott, Stephanie J. Valberg.

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
