## [Decision Letter · Decision Letter 0]

11 Nov 2020

PONE-D-20-29481

Pathways of calcium regulation, electron transport, and mitochondrial protein translation are molecular signatures of susceptibility to recurrent exertional rhabdomyolysis in Thoroughbred racehorses

PLOS ONE

Dear Dr. Valberg,

Thank you for submitting your manuscript to PLOS ONE. After careful consideration, we feel that it has merit but does not fully meet PLOS ONE’s publication criteria as it currently stands. Therefore, we invite you to submit a revised version of the manuscript that addresses the points raised during the review process.

Please, consider all the variables involved in the experimental protocol to have a better assessment of the conclusions strength.

We look forward to receiving your revised manuscript.

Kind regards,

Agustín Guerrero-Hernandez

Academic Editor

PLOS ONE

Journal Requirements:

2. In your Methods section, please state the volume of the blood samples collected for use in your study.

3. We note that you are reporting an analysis of a microarray, next-generation sequencing, or deep sequencing data set. PLOS requires that authors comply with field-specific standards for preparation, recording, and deposition of data in repositories appropriate to their field. Please upload these data to a stable, public repository (such as ArrayExpress, Gene Expression Omnibus (GEO), DNA Data Bank of Japan (DDBJ), NCBI GenBank, NCBI Sequence Read Archive, or EMBL Nucleotide Sequence Database (ENA)). In your revised cover letter, please provide the relevant accession numbers that may be used to access these data. For a full list of recommended repositories, see http://journals.plos.org/plosone/s/data-availability#loc-omics or http://journals.plos.org/plosone/s/data-availability#loc-sequencing.

"The authors have declared that no competing interests exist.".

We note that one or more of the authors are employed by a commercial company: Equine Integrated Medicine, PLC'

Reviewers' comments:

Reviewer's Responses to Questions

**Comments to the Author**

1. Is the manuscript technically sound, and do the data support the conclusions?

Reviewer #1: Yes

Reviewer #2: Yes

2. Has the statistical analysis been performed appropriately and rigorously? 

Reviewer #1: Yes

Reviewer #2: I Don't Know

3. Have the authors made all data underlying the findings in their manuscript fully available?

Reviewer #1: Yes

Reviewer #2: Yes

4. Is the manuscript presented in an intelligible fashion and written in standard English?

Reviewer #1: Yes

Reviewer #2: Yes

5. Review Comments to the Author

Reviewer #1: The search for environmental factors and a genetic predisposition which impact the expression of RER in the horse is of great interest in the practice of sports medicine of this species. The present study has sought the differential expression patterns of specific genes/proteins and GO pathways in RER-susceptible horses. The Authors propose a scenario whereby horses susceptible to RER have enhanced storage of Ca2+ in the sarcoplasmic reticulum. Based on these main results, the authors concluded that it enhances muscle force and energy metabolism within a physiologic range of myoplasmic Ca2+. It is very important but not new finding. However the unique is that the newest and more sophisticated research techniques were used. In addition, this is the first study where RER-susceptible horses were used. In addition, the horses affected by dantrolene were used. The main limitation of the study is low number of horses and different age of them. Thus, the training level of the horses vary.

Specific comments

L45- side effects of dantrolene should be mentioned.

L77 –what was their training regimen? Were their trained according to the same protocol? What was the exertion after which the samples were obtained? Please specify.

L78 –“Race-13”, it is not clear what it means.

L81- how veterinarian confirmed RER? It was after trotting test or regular training session? Was the LDH measured?

L84 – the number of horses in Untreated RER-susceptible and treated group should be mentioned also in those paragraph.

L96 – “Samples were obtained in the morning between 1 and 5.5 h after either jogging or galloping exercise on the track” – what do You mean jogging? It is confusing if the horses were exercised or not in this study? Please specify the training session after which the samples were obtained.

L243 – whereas there were no statistical differences, there is a huge difference between 2 years old and almost 6 years old race horse during training. The Authors should take it into consideration.

Table 2 - Adjusted p value is hard to follow.

L499 – what do Authors mean by “tight control”? Expand Your mind.

L 535 – The statement is risky. Because during chronic stress it was demonstrated that cortisol release is lower. In addition, it should be taken in to consideration that cortisol plays an essential role in adaptation during exercise. Upregulation of this hormone leads to increase of protein synthesis It can also increase glycogen deposition and storage in the liver.

L 595 – the time of sampling should be the same in next study because in 5,5h after exercise the proteins may be degraded.

L599 – the timing of samples collection is crucial because of as a result of the vasoconstrictor and vasodilatory response after the exercise. Thus, the blood influx in to the muscles may vary.

Reviewer #2: this manuscript is considered new in the field of recurrent exertional rhabdomyolysis in Thoroughbred racehorses

and includes a lot of work and its goals are multiple

It is clear that the abstract was written in a hurry and it needs to be revised

the role of the serum CPK and AST in the diagnosis of exertional rhabdomyolysis should be focused in the text

6. PLOS authors have the option to publish the peer review history of their article (what does this mean?). If published, this will include your full peer review and any attached files.

Reviewer #1: No

Reviewer #2: No

---

## [Author Response · Author response to Decision Letter 0]

2 Dec 2020

Editors comments

1) We changed the methods section to indicate 7 ml of blood was collected

2) We did upload the data and this was indicated in the paper. I have highlighted the information on where data can be found is in the paper to show where that is located.

3) We would like to amend the funding statement to state. 

The funding agency provided funds to perform the research study and partial salary for one author (DV) but did not have any additional role in the study design, data collection and analysis, decision to publish, or preparation of the manuscript

4) We would like to amend the competing interest statement. 

One of the authors (CF) is a practicing veterinarian and owner of the business Equine Integrated Medicine PLC. The business did not play a role in the funding or design of the study and Dr. Fenger did not receive any financial remuneration for participating in the study. Dr. Fenger identified horses in her practice that were healthy controls or susceptible to RER and facilitated biopsy samples that were obtained by another author (SV). Dr. Fenger was involved in reviewing the data regarding accuracy of the horse’s clinical phenotypes and approving the manuscript for publication. 

There are no other relevant declarations relating to employment, consultancy, patents, products in development, or marketed products.

Response to reviewers

We thank the reviewers for their time and insights into our manuscript. 

Reviewer #1: The search for environmental factors and a genetic predisposition which impact the expression of RER in the horse is of great interest in the practice of sports medicine of this species. The present study has sought the differential expression patterns of specific genes/proteins and GO pathways in RER-susceptible horses. The Authors propose a scenario whereby horses susceptible to RER have enhanced storage of Ca2+ in the sarcoplasmic reticulum. Based on these main results, the authors concluded that it enhances muscle force and energy metabolism within a physiologic range of myoplasmic Ca2+. It is very important but not new finding. However the unique is that the newest and more sophisticated research techniques were used. In addition, this is the first study where RER-susceptible horses were used. In addition, the horses affected by dantrolene were used. The main limitation of the study is low number of horses and different age of them. Thus, the training level of the horses vary. 

Specific comments 

L45- side effects of dantrolene should be mentioned.- 

This has been added. We are only aware of side effects in horses under anesthesia. Line 57 

L77 –what was their training regimen? Were their trained according to the same protocol? What was the exertion after which the samples were obtained? Please specify. 

The training regime was similar and has been added Line 96. The exercise preceding the biopsy was similar between groups and has been added Line 158 

L78 –“Race-13”, it is not clear what it means. 

This is the brand of horse feed which is further explained now Line 94 

L81- how veterinarian confirmed RER? It was after trotting test or regular training session? Was the LDH measured? 

The AST and CK were measured when a horse had clinical signs of exertional rhabdomyolysis (pain, stiffness, sweating, reluctance to move). This has been added. Line 102. We do not routinely analyze LDH in muscle cases because it is not specific for muscle and because AST provides the needed additional information on the chronicity of rhabdomyolysis. 

L84 – the number of horses in Untreated RER-susceptible and treated group should be mentioned also in those paragraph. 

added 

L96 – “Samples were obtained in the morning between 1 and 5.5 h after either jogging or galloping exercise on the track” 

– what do You mean jogging? This is the term used at the racetrack for trotting, this has been changed to trotting in the manuscript Line 96, Line 159 

It is confusing if the horses were exercised or not in this study? Please specify the training session after which the samples were obtained. 

Added Line 159 

L243 – whereas there were no statistical differences, there is a huge difference between 2 years old and almost 6 years old racehorse during training. The Authors should take it into consideration. 

We have acknowledged that ideally horses would have been the same age in the discussion line 695. We did not have a significant difference in age between the control and affected groups and all horses, including two-year-olds, were in full race training. Older horses would be expected to have more mitochondrial proteins whereas we found that the RER group which included the one 6 year old had fewer mitochondrial proteins. 

Table 2 - Adjusted p value is hard to follow. 

The DEG data had adjusted p-values, the p-values for DEP were not adjusted p-values. The adjusted p-values are q-values, a corrected p-value based on estimations for the proportion of false positives. Significance was determined using a false discovery rate (FDR) ≤ 0.05 to correct for multiple testing in both DEG and DEP. The p-values for proteomics were determined through permutation, the cutoff equivalent to an FDR ≤ 0.05 is a p-value ≤ 0.01 (for this analysis). 

Line 220. Multiple test correction was performed with false discovery rate less than 0.05. q-values, an adjusted or corrected p-value that is based on estimations of the proportion of false positives, were used in tables to show significance.

L499 – what do Authors mean by “tight control”? Expand Your mind. We reworded this to make the sentence clearer. 

Line 581 Our results suggest that regulation of myoplasmic Ca2+homeostasis is perturbed in horses susceptible to- but not currently experiencing RER- when housed in a race-training environment. Marked alterations in the expression of mitochondrial proteins in RER susceptible horses could be related to the role of mitochondria in buffering myoplasmic Ca2+ which could have a detrimental down-stream effect on the mitochondrial respiratory chain and apoptosis [39]. 

L 535 – The statement is risky. Because during chronic stress it was demonstrated that cortisol release is lower. In addition, it should be taken in to consideration that cortisol plays an essential role in adaptation during exercise. Upregulation of this hormone leads to increase of protein synthesis It can also increase glycogen deposition and storage in the liver. 

We have changed this to indicate that “Conditions of chronic stress with associated prolonged beta-adrenergic stimulation could produce post-translational modification of RYR1 in young fillies that trigger enhanced Ca2+ release in RER-susceptible horses.” Line 623 We did not mean to imply that cortisol was detrimental but meant that cortisol levels were an indicator in RER horses that they could be undergoing chronic stress. The beta-adrenergic stimulation of CALM2 and its effect on hyperphosphorylation of RYR1 could be a reason why stress impacts the expression of RER in a fashion similar to what has been demonstrated with chronic heart failure. 

L 595 – the time of sampling should be the same in next study because in 5,5h after exercise the proteins may be degraded. 

Yes this would definitely be ideal to have been able to make this all the same. 

L599 – the timing of samples collection is crucial because of as a result of the vasoconstrictor and vasodilatory response after the exercise. Thus, the blood influx in to the muscles may vary. This is an excellent point. Fortunately, there was not a significant difference in the timing of the sample relative to exercise across the three groups. We have add post exercise blood flow as a fact Line 689. 

Reviewer #2: this manuscript is considered new in the field of recurrent exertional rhabdomyolysis in Thoroughbred racehorses and includes a lot of work and its goals are multiple 

It is clear that the abstract was written in a hurry and it needs to be revised 

We have revised the abstract. 

The role of the serum CPK and AST in the diagnosis of exertional rhabdomyolysis should be focused in the text 

We have added additional information on the way the horses were diagnosed with RER Line 102. Horses were diagnosed with RER based on both clinical signs and on the presence of abnormal elevations in CK and AST at the time clinical signs were present.

---

## [Decision Letter · Decision Letter 1]

14 Dec 2020

Pathways of calcium regulation, electron transport, and mitochondrial protein translation are molecular signatures of susceptibility to recurrent exertional rhabdomyolysis in Thoroughbred racehorses

PONE-D-20-29481R1

Dear Dr. Valberg,

We’re pleased to inform you that your manuscript has been judged scientifically suitable for publication and will be formally accepted for publication once it meets all outstanding technical requirements.

Kind regards,

Agustín Guerrero-Hernandez

Academic Editor

PLOS ONE

Additional Editor Comments (optional):

Reviewers' comments:

Reviewer's Responses to Questions

**Comments to the Author**

1. If the authors have adequately addressed your comments raised in a previous round of review and you feel that this manuscript is now acceptable for publication, you may indicate that here to bypass the “Comments to the Author” section, enter your conflict of interest statement in the “Confidential to Editor” section, and submit your "Accept" recommendation.

Reviewer #1: All comments have been addressed

Reviewer #2: All comments have been addressed

2. Is the manuscript technically sound, and do the data support the conclusions?

Reviewer #1: Yes

Reviewer #2: Yes

3. Has the statistical analysis been performed appropriately and rigorously? 

Reviewer #1: Yes

Reviewer #2: I Don't Know

4. Have the authors made all data underlying the findings in their manuscript fully available?

Reviewer #1: Yes

Reviewer #2: (No Response)

5. Is the manuscript presented in an intelligible fashion and written in standard English?

Reviewer #1: Yes

Reviewer #2: Yes

6. Review Comments to the Author

Reviewer #1: Dear Authors,

Now I have no doubts about an article. It is important and interesting topic according to muscle failure in the race horses.

Reviewer #2: This manuscript reported that recurrent exertional rhabdomyolysis in Thoroughbred racehorses is associated with alterations in proteins, genes and pathways impacting myoplasmic Ca2+ regulation, the mitochondrion and protein degradation with opposing effects on mitochondrial transcriptional/translational responses and mitochondrial protein content. Great care has been observed to improve the work in the revised manuscript.

7. PLOS authors have the option to publish the peer review history of their article (what does this mean?). If published, this will include your full peer review and any attached files.

Reviewer #1: No

Reviewer #2: No

---

## [Editor Report · Acceptance letter]

20 Jan 2021

PONE-D-20-29481R1 

Pathways of calcium regulation, electron transport, and mitochondrial protein translation are molecular signatures of susceptibility to recurrent exertional rhabdomyolysis in Thoroughbred racehorses 

Dear Dr. Valberg:

I'm pleased to inform you that your manuscript has been deemed suitable for publication in PLOS ONE. Congratulations! Your manuscript is now with our production department. 

Kind regards, 

on behalf of

Dr. Agustín Guerrero-Hernandez 

Academic Editor

PLOS ONE